# Prevalence and Predictors of Per- and Polyfluoroalkyl Substances (PFAS) Serum Levels among Members of a Suburban US Volunteer Fire Department

**DOI:** 10.3390/ijerph18073730

**Published:** 2021-04-02

**Authors:** Judith M. Graber, Taylor M. Black, Nimit N. Shah, Alberto J. Caban-Martinez, Shou-en Lu, Troy Brancard, Chang Ho Yu, Mary E. Turyk, Kathleen Black, Michael B. Steinberg, Zhihua Fan, Jefferey L. Burgess

**Affiliations:** 1Department of Biostatistics and Epidemiology, Rutgers School of Public Health, Rutgers the State University of New Jersey, New Brunswick, NJ 08854, USA; nns52@sph.rutgers.edu (N.N.S.); sl1020@sph.rutgers.edu (S.-e.L.); brancardte@gmail.com (T.B.); 2Rutgers Environmental and Occupational Health Sciences Institute, Rutgers the State University of New Jersey, New Brunswick, NJ 08854, USA; tmb220@eohsi.rutgers.edu (T.M.B.); kgb3@eohsi.rutgers.edu (K.B.); 3Sylvester Comprehensive Cancer Center, University of Miami Miller School of Medicine, Miami, FL 33136, USA; ACaban@med.miami.edu; 4Environmental and Chemical Laboratory Services, New Jersey Department of Health, Trenton, NJ 08628, USA; Chang.Yu@doh.nj.gov (C.H.Y.); tina.fan@doh.nj.gov (Z.F.); 5School of Public Health, University of Illinois, Chicago, IL 60612, USA; mturyk1@uic.edu; 6Department of Medicine, Rutgers Robert Wood Johnson Medical School, New Brunswick, NJ 08903, USA; steinbmb@rwjms.rutgers.edu; 7Mel and Enid Zuckerman College of Public Health, University of Arizona, Tucson, AZ 85724, USA; jburgess@arizona.edu

**Keywords:** per-and polyfluoroalkyl substances (PFAS), volunteer firefighters, PFAS prevalence

## Abstract

Background: Per-and polyfluoroalkyl substances (PFAS), are ubiquitous pollutants associated with adverse health outcomes. High PFAS levels have been demonstrated among career firefighters; less is known about PFAS levels among volunteer firefighters who comprise two-thirds of US firefighters. Methods: Volunteer fire department members completed a survey and provided blood samples. We calculated geometric means and 95% CIs for PFAS reported by the National Health and Nutrition Examination Survey (NHANES). We compared PFAS distribution and levels among non-Hispanic white adult male study participants to those in the 2015–2016 and 2017–2018 NHANES cycles. We assessed associations between PFAS serum levels and years of firefighting controlling demographics and occupation using linear regression. Results: Participant’s average age was 46.6 years (sd. 17.1). Perfluorododecanoic acid (PFDoA) was detected in almost half study but <3% of NHANES participants; serum levels of PFDoA, perfluorononanoic acid (PFNA), and perfluorodecanoic acid (PFDA) were elevated among participants compared with NHANES. Serum levels of both PFDA and PFDoA were positively associated with years of firefighting. Conclusions: Volunteer firefighters may have a different serum profile and levels of PFAS than the general population. Future work in this area should include volunteer firefighters from other geographic locations and assess sources of PFAS exposure.

## 1. Introduction

Per- and polyfluoroalkyl substances (PFAS), also known as ‘forever chemicals’, are a global environmental and health concern due to their ubiquitous presence in the environment, tendency to bioaccumulate, and the growing evidence of adverse human health effects at very low levels of exposure [1]. They are synthetic, thermally stable compounds with unique non-stick surfactant properties [2]. As such, PFAS are widely used in consumer products including food packaging, cookware coatings, water-resistant products such as cookware, dental floss, and in-home furniture and carpeting [1]. PFAS are also used for many industrial processes. Human PFAS exposure can occur through ingestion of contaminated food or water, inhalation, and there is evidence from rodent models of dermal adsorption [3]. Contamination of ground water has been commonly reported around airports and military bases, because PFAS (primarily perfluorooctanesulfonic acid (PFOS) and perfluorooctanoicacid (PFOA)) have historically constituted one to five percent of Class B aqueous film-forming foams (AFFF) that are used for fire suppression in those settings [4]. Other industrial point sources of community ground water have been reported [5,6,7]. 

In humans, the serum half-lives perfluorohexanesulfonic acid (PFHxS), PFOS, and PFOA were reposted by one study as 5.3, 3.4, and 2.7 years, respectively [8]. They are readily transported and have been detected in ecosystems from the Arctic to the Antarctic [9,10]. Almost all U.S. residents have detectable levels of one or more of the most studied long-chain PFAS: PFOA, PFOS, PFHxS, and perfluorononanoic acid (PFNA). PFAS exposure has been associated with multiple adverse human health outcomes including dyslipidemia, cardiovascular disease, immune suppression, kidney disease, and endocrine disruption [1]. There is evidence that several long-carbon chain PFAS may be carcinogenic [11]. In 2015 the International Agency for Research on Cancer classified PFOA as a group 2B (possible) carcinogen for kidney and testicular cancers [12]. 

Firefighters can be exposed to PFAS through multiple pathways. Firefighters’ protective clothing (aka, gear or turnout gear) was historically treated with PFAS to provide water and stain resistance properties; evidence of PFAS in all layers of gear has been reported [13]. Residential, commercial, and industrial building structure and vehicle fires may burn products that contain PFAS including electronics, furniture, carpeting, and insulation and release particles that can be inhaled or settle on gear and skin [14]. Firefighters who use AFFF have been shown to have increased serum concentrations of PFOS and PFHxS that was positively associated with years of firefighting [15]. 

Higher mean serum levels of some PFAS, including PFOA, PFOS, PFHxS, PFNA, and perfluorodecanoic acid (PFDA) have been observed among firefighters than those of demographically similar subsets of the general population [16]. Most research assessing PFAS exposure in firefighters has been conducted among career (paid) firefighters. However, two-thirds (67%) of firefighters in the US serve as volunteers [17]. Volunteer firefighters perform the same tasks as their career counterparts, but often with less protection and risk reduction. 

There is a significant gap in our understanding of PFAS exposure in US firefighters and specifically, among volunteer firefighters. To begin to address this gap, we conducted a biomonitoring study in a large suburban volunteer fire department with the primary goal of assessing the distribution and levels of PFAS compounds detected in serum compared to the general US population as represented by the National Health and Nutrition Examination Survey (NHANES). NHANES is a population-based survey that publishes data in two-year cycles. When this study begun, the most recent cycle for which NHANES PFAS serum levels were publicly available was 2015–2016, and we completed our comparisons using those data. However, before the study was concluded PFAS data from the 2017–2018 NHANES cycle became publicly available. Serum concentration of some long chain PFAS are declining in the general population, including PFOS, PFOA and PFNA, [18] (which have previously been reported as elevated in biomonitoring studies of firefighters). The publishing of the 2017–2018 presented an opportunity to add a secondary aim to the study: To assess whether a comparison with the NHANES data collected closer in time to our study data (2019) would alter our findings or interpretation. As such, we compared the distribution and levels of PFAS compounds among volunteer firefighter study participants to those of the two most recent NHANES cycles. Within the volunteer firefighters, we also explored associations between firefighting exposures and PFAS serum levels.

## 2. Materials and Methods

### 2.1. Collaboration

This study was a collaborative effort between the New Jersey Firefighter Cancer Assessment and Prevention Study (CAPS), the US Firefighter Cancer Cohort Study (FFCCS), and the New Jersey Department of Health, Public Health and Environmental Laboratories (NJDOH-PHEL) biomonitoring project. CAPS was a two-year funded project designed to develop and launch a long-term research infrastructure to understand and prevent cancer among the over 35,000 active NJ firefighters, 80% of whom are volunteers [19]. CAPS collaborates with and uses the methodology of the FFCCS, which was established in 2016 and provides a national research framework to collect and integrate firefighter epidemiologic surveys, biomarkers and exposure data focused on carcinogenic exposures and health effects. For this project, CAPS used the FFCCS Annual Cancer Survey, after adapting it for use by volunteer firefighters (see below section, “Data Collection: The Survey”). Biospecimen collection and processing procedures were closely coordinated with the NJDOH-PHEL, who also analyzed participant PFAS serum levels. All collaborators participated in the interpretation of the biomonitoring results. 

### 2.2. Inclusion Criteria and Enrollment Procedures

A cross-sectional study of new recruits and incumbent members of a large suburban volunteer fire department was conducted in 2019. Preceding enrollment, the study team held informational sessions with fire department leadership and membership. Study enrollment and data collection sessions were held during regularly scheduled weekly training at the department training center or fire stations. Eligible participants were at least 18 years old and were new recruits, incumbent or former members of the volunteer fire department. At each of six enrollment sessions, study personnel administered informed consent. Consented participants completed an online survey and were asked to provide a blood and urine sample. 

### 2.3. Data Collection: The Survey

The survey was administered electronically using REDCap, a secure online data collection and management system [20]. Participants had the option of completing it on a study-provided laptop or using their own mobile device. The FFCCS survey assesses cancer risk factors, screening behaviors, health care access and utilization, occupational exposures, and firefighting experience. At the time of this study, the survey was written with administration to career firefighters in mind. The CAPS team adapted the survey questions on occupational history and firefighting service so they were applicable to volunteer firefighters. The survey took approximately 45 min to complete. 

### 2.4. Specimen Collection and Laboratory Analysis

Blood was collected in 10 mL serum separator tube (SST) tubes and stored on ice through transport to Rutgers University (RU) where they were processed. Serum was stored at −30 °C until transported on dry ice to the NJDOH-PHEL facility for PFAS analysis. Sera were analyzed by PHEL-Environmental Chemical Laboratory Services (ECLS) using the optimized PFAS testing method, NJDOH PFAS method. This method was optimized from the U.S. Centers for Disease Control and Prevention (CDC) method (#6304.04) as part of the New Jersey State Biomonitoring Program and has been employed to measure PFAS levels for more than a thousand de-identified human sera in a statewide PFAS biomonitoring project [21]. The method utilizes a high-throughput online solid phase extraction (SPE) (Spark Holland, Emmen, Netherlands) system and a highly sensitive tandem mass spectrometer (MS/MS) (Sciex QTrap 6500, Framingham, MA, USA), which can detect 12 target PFAS analytes as low as <0.1 ng/mL within a 10-min running time. Detailed method development and optimizations were described, respectively [22]. The ECLS laboratory has participated in the CDC’s bi-annual Biomonitoring Quality Assurance Support Program (BQASP) for PFAS in serum since 2016 and has passed all proficiency tests by scoring 100% score. Additionally, the ECLS laboratory quality assurance/quality control procedures include that all analyzed samples and concentration data are systematically reviewed and audited. Prior to reporting results, all concentration data are validated against data quality control protocols.

### 2.5. Study Population—National Health and Nutrition Examination Survey (NHANES)

The comparison population for the study was drawn from participants of the 2015–2016 and 2017–2018 NHANES cycles. NHANES is an ongoing cross-sectional survey that is administered by the CDC. NHANES participants are selected using a multistage cluster sample design to be representative of the US non-institutionalized population. Blood is collected from participants 12 years and older. Serum from one-third of NHANES serum samples were analyzed for a panel of 10 PFAS compounds in the 2015–2016 and 2017–2018 cycles. NHANES releases publicly available data with participant demographic and laboratory results, and tools for analysis of the sample-survey data structure. NHANES documentation incudes detailed descriptions of the laboratory methods [23].

### 2.6. Statistical Analysis

#### 2.6.1. Demographics and Firefighting Characteristics 

Demographics characteristics for CAPS and NHANES participants were estimated using survey responses and included:Age: defined as years between date of birth and survey date; categorized into approximately equal quartiles of 18–34, 35–49, 50–59, and 60 or more for bivariate analysisSex: categorized into male/femaleRace/ethnicity: categorized into non-Hispanic white/other because the majority of CAPS participants were non-Hispanic whiteEducation attainment: defined for the highest level of education achieved as high school graduate; some college/Associates degree; or 4-year college degree or moreOccupation: categorized as construction/manufacturing, government/clerical, service provider, and other occupation.Firefighting characteristics for CAPS participants included:-Ever employed (for pay) as a firefighter: categorized as current, former, or never having been a paid firefighter-Years of firefighting service: estimated as the difference between the first year of fire department membership (volunteer or career, whichever was earlier) and either the last year of active department membership or the survey date), and categorized into approximately equal quartiles for bivariate analysis (0 to 5, 6 to 19, 20 to 34, and 35 or more years)-Firefighting calls: Calculated as the cumulative number of firefighting calls responded to over the total years serving as a firefighter and standardized to one year; categorized into approximately equal quartiles of 0 to 4, 5 to 9, 10 to 19, and 20 or more calls per year.

#### 2.6.2. Comparison of PFAS Levels between CAPS and NHANES Participants

As the CAPS participants were 92% male and 90% non-Hispanic white, we restricted the comparison with NHANES participants to non-Hispanic white males ages 18 to 79. Differences between serum concentrations were assessed for the eight PFAS compounds assessed by both NJDOH-PHEL and NHANES and for which at least one study participant had a serum concentration above the limit of detection (LOD). For NHANES the LOD was 0.10 for each compound. Samples below the LOD were assigned an imputed value of the limit of detection divided by the square root of 2 (0.07 ng/mL). NJDOH assigned a different lower limit of detection for each PFAS compound, all of which were lower than those of NHANES. To ensure a more accurate comparison with NHANES, all NJDOH values were assigned the same limit of detection as NHANES (0.10 ng/mL). Inconsistencies for PFAS acronyms used by NJDOH and NHANES were resolved by communication with an NJDOH scientist. NHANES acronyms are used hereafter. PFOA and PFOS were analyzed as the sum of their respective linear and branched-chain polymers.

PFAS prevalence was defined as the number of detects above the LOD divided by the number of samples for each PFAS within each study group. Geometric mean serum levels and corresponding 95% confidence intervals were calculated for each of the eight compounds. Geometric means and 95% confidence intervals for NHANES were calculated using the stratum, cluster, and subsample weights provided by NHANES; subsamples were analyzed using domains. 

#### 2.6.3. Association between Firefighting Experience and PFAS Levels (CAPS Participants Only)

An a priori decision was made to assess associations between years of firefighting experience and any PFAS serum levels found to have significantly higher geometric mean serum concentrations in CAPS compared to NHANES participants. This analysis included all 135 CAPS participants who provided a blood sample. Due to the non-normal distributions of serum concentrations for three of the PFAS compounds (PFNA, PFDA and perfluorododecanoic acid (PFDoA)), non-parametric approaches were used to assess bivariate associations between PFAS and firefighting experience. Correlations between each of the three PFAS and the continuous variables including age (in years), cumulative years of firefighting, and firefighting calls (number per year) were calculated using Spearman correlation coefficients and *p*-values. Associations between each PFAS and categorical variables (primary occupation, education level, and ever been a career firefighter) were assessed by calculating geometric means and 95% confidence intervals and *p*-values for differences between levels for each variable. 

Associations between serum concentrations of log-transformed PFNA, PFDA and PFDoA and firefighting characteristics were assessed using generalized linear regression models, one for each outcome. Each model included firefighting year, and age modeled in their continuous (log-linear) format. For ease of interpretation, the percent change in the exposure to each PFAS is presented as an increase in 10 years of firefighting experience. Categorical covariates included ever career firefighter (reference = never); educational level (reference = high school graduate or less) and primary occupation (reference = service provider). 

#### 2.6.4. Sensitivity Analysis

A sensitivity analysis was conducted to compare associations between firefighting experience and PFAS levels for current firefighters with and without experience as a career firefighter. These models also included calls per year. The association between firefighting experience and PFAS levels for CAPS firefighters with no career experience was modeled using generalized linear regression models as described above. All statistical analyses were performed in SAS/STAT statistical software (SAS Inc, Cary, NC, USA).

## 3. Results

### 3.1. Participant Characteristics

While 138 members of the volunteer fire department enrolled in CAPS, two were missing demographic data and one did not have blood drawn, so this study included 135 participants. The majority of enrollees were male (95%) and non-Hispanic white (90%). The mean age was 47 years old and the average years of firefighting experience was 20. Almost two thirds had some college education (72%). Less than one-fifth of the firefighters had ever worked as a career firefighter (18%) (Table 1).

### 3.2. Comparison of the Distribution of Detected PFAS Compounds

The distribution of PFOA, PFOS and PHFxS was similar between the non-Hispanic White male volunteer firefighters in our study and demographically similar NHANES participants. However, the distribution of the other measured PFAS varied between CAPS and the NHANES participants (Table 2). While the proportion of NHANES participants with detectable levels of PFNA appeared to decline from the NHANES 2015–2016 to 2017–2018 cycles (from 98% to 92%), PFNA was detected in serum of all CAPS study participants. Perfluorododecanoic acid (PFDoA) was detected in 80% of CAPS participants, but <3% of NHANES participants in the 2015 to 2016 cycle, and no participants in the 2017–2018 NHANES cycle. 

### 3.3. Comparison of Mean Serum PFAS Concentrations

When comparing CAPS participants to NHANES 2015–2016 and 2017–2018 cycles, the serum levels fluctuated by compound but the overall relationships were similar (Table 2). Results comparing CAPS study participants to the NHANES 2017–2018 participants are summarized here with the exception of PFDoA, for which there were no detected levels among the NHANES 2017–2018 participants. There were significant differences in mean serum concentrations for five of the eight PFAS. CAPS members had significantly lower geometric mean serum concentrations compared to NHANES for PFOS (4.25 ng/mL and 6.08 ng/mL, respectively; −43%) and MeFOSAA (0.08 ng/mL and 0.15 ng/mL, respectively; −88%). CAPS participants had significantly higher mean serum concentrations compared to NHANES of PFNA (0.97 ng/mL and 0.46 ng/mL, respectively; +53%), PFDA (0.31 ng/mL and 0.19 ng/mL, respectively; +39%), and PFDoA (0.14 ng/mL and 0.07 ng/mL [NHANES 2015–2016]; +50%).

### 3.4. Association between Firefighting Experience and PFAS Levels (CAPS Participants Only)

PFNA serum concentration was significantly correlated with age, years of firefighting and yearly number of calls responded to in bivariate analysis (Table 3), however after controlling for age, educational level and occupation these associations were no longer significant. Higher serum levels of both PFDA and PFDoA were positively and significantly associated with years of firefighting service after controlling for age, educational level, and occupation. For every increase of 10 years of firefighting, the expected value of PFDA increased by 8% (95% CI: 1%, 15%) and the expected value of PFDoA increased by 19% (95% CI: 9%, 30%). PFDoA serum levels were also positively associated with never, compared to ever having been a paid firefighter (*p*-value 0.026). Unexpectedly, having any college education compared to none was positively associated with increased serum levels of both PFDA and PFDoA (Table 4).

In the sensitivity analysis assessing the association between firefighting experience with PFAS serum-concentration of firefighters among study participants with no career firefighting experience (only experience only in volunteer fire departments), adjusting for demographic factors, the associations between PFDA and PFDoA with years of firefighting service were attenuated (Table 5). 

## 4. Discussion

We found that the prevalence and level of PFAS chemicals in serum differed between members of a large suburban volunteer fire department and the general US population, including that PFDoA was detected in 80% of study subjects but in none of the NHANES participants in the 2017–2018 cycle. As well, we observed significantly higher serum levels of PFNA, PFDA, and PFDoA, but lower levels of PFOS and MeFOSAA, than NHANES participants of the same age, gender, and race-ethnicity (adult, non-Hispanic white males). These findings are generally consistent with prior research in that firefighters have elevated serum levels of some long-chain PFAS when compared to the general population, including PFHxS, PFDA and PFNA. As well, some firefighters with history of using AFFF have elevated serum levels of PFOA and PFOS. However, much of the prior research on PFAS exposure among firefighters has been conducted among career firefighters. To our knowledge this is the first study of its size to evaluate PFAS exposure among volunteer firefighters. 

In our side by side comparison of our study participants enrolled in 2019 to the two most recent NHANES cycles, 2015–2016 and 2017–2018, the overall interpretation of our findings would have been similar had we used the earlier NHANES cycle (2015–2016). However, because concentrations of PFAS are dynamic in the general population, with some average levels declining and some increasing, reliance on the 2015–2016 rather than the 2017–2018 data which is more temporally proximal to the 2019 CAPS study, would have had introduced errors in estimating differences between the prevalence and serum levels of some PFAS among our two study populations.

Our observation that mean PFDA serum concentrations were significantly elevated in these volunteer firefighters compared to NHANES participants is notable. It is consistent with a 2015 biomonitoring study of 101 California career firefighters that reported their PFDA levels were three times higher than those in NHANES participants. Of note, the serum concentrations of PFDA levels observed in the California firefighters (and the 2011–2012 NHANES sample 0.90 μg/L; 95% CI, 0.78 to 1.03), were much higher than those we observed in this study [24]. This may reflect that serum levels of some long-chain PFAS are falling as they are phased out of industrial use and consumer products due to health and environmental concerns. These career firefighters in California did not have detectable serum levels of PFDoA; in contrast the volunteer firefighters from New Jersey had a high prevalence (80%) and significantly elevated serum levels of PFDoA, with the geometric mean twice as high as in the NHANES participants. The source of PFDoA exposure in these volunteer firefighters is an important direction for future investigation. While these firefighters differed from those in the California study by both working structure (volunteer vs. career) and geography, it is noteworthy that in our study PFDA and PFDoA serum levels were positively correlated with both years of being a firefighter and never having worked as career firefighter. However, community exposure may also be contributing to PFAS exposures in the study participants as the area in New Jersey had a history of industrial pollution from dye manufactures and other industries which may use PFAS in their manufacturing processes [25]. In a 2020 study of PFAS in surface water and fish conducted by sampling 11 waterways around New Jersey based on proximity to potential sources of PFAS in recreational areas. Four of the sampling areas were adjacent to the township where the current study was conducted because they are within the drainage basin of the Joint Base Maguire-Dix-Lakehurst complex. Of the water samples adjacent to the study town, all contained detectable levels of PFOA and PFOS but not of PFDoA or PFNA; PFNA was detected in one location [26]. These results do not lend support to an environmental source of PFDoA in these firefighters.

We observed elevated serum concentrations of PFNA in our study participants, which was not associated with years of firefighting after controlling for age, occupation, and educational level. Other biomonitoring studies have observed elevated levels of PFNA, as well as PFHxS and PFOS among firefighters who have worked with AFFF [15]. However, the volunteer fire department whose members participated in this study rarely used AFFF but some of the participants may have done so in previous departments or in the armed services, however we did not see an association between being a career firefighter an PFAS levels serum levels or with previous military service (data not shown). There are some areas of New Jersey with uniquely high PFNA ground water contamination, [5] as well some studies in the general population that found diet can account for over half of PFNA exposure [27,28]. As such the lack of association with firefighting experience may point to an environmental rather than firefighting-related source. Including community members with a similar socio-demographic profile but no firefighting experience in future studies of PFAS exposure could offer insights into the important question of PFAS exposure sources.

A biomonitoring study of New Jersey residents conducted using remnants of laboratory specimen and blood donations acquired between 2016 and 2018 and used post-sample stratified weights to estimate population parameters reported a somewhat different distribution of PFAS in adult New Jersey males than was seen among thee volunteer firefighters. They observed lower geometric levels of PFNA (0.88 ng/mL, 95% CI: 0.77, 1.01] and PFDA (0.23 ng/mL, 95% CI: 0.18, 0.29], although in both cases the confidence intervals overlapped [21]. 

The strong and consistent positive association observed between having any college education and serum PFDA and PFDoA concentrations was unexpected. We were unable to evaluate variation in household income in this study, but education is typically strongly correlated with socioeconomic status and associated lifestyle factors. Studies of dietary contribution to PFAS body burden find that the major contributors are fish, meat, fruit and eggs [29]. Higher intake of fish and fruit are associated with higher income in the US and so this may explain part of the observed association. As well, PFAS are used on furnishings and in carpeting to make them stain resistant, so higher serum levels may reflect consumer product patterns associated with income. This is an important area for future research.

Our study had a number of notable strengths including being among the first study of its size to investigate PFAS exposure among volunteer firefighters. In New Jersey where this study was conducted, more than 80 percent of the approximately 37,000 firefighters are volunteers. In the US, the percentage of US firefighters who are volunteers is increasing while the number of career freighters is decreasing [17]. Volunteer firefighters train for and perform the same tasks as career firefighters, but often with less protection and risk reduction. They are always on-call, and so could potentially accumulate more years of firefighting -related exposures than their career counterparts [30]. Compared with career firefighters, significantly more US volunteer firefighters are females: 4 vs. 11%. We were unable to find reliable statistics on the proportion of US volunteer firefighters who are non-Hispanic white. In the career service, approximately 8% are African American, 8% Hispanic and 1 percent Asian or other. Future studies of volunteer firefighters should attempt to include more females and people of color as their occupational and exposure profile may differ.

Another strength of this study was that it was a scientific collaboration with the national FFCCS [31]. We used their Annual Cancer Survey, an instrument that had already been used extensively within firefighter populations. However, a limitation inherent in capturing firefighting history by survey is obtaining an accurate exposure assessment for firefighters given the diversity in the types of calls they respond to, their variable call volume, and often their long duration of firefighting experience. The inability to precisely capture firefighters’ exposures may result in exposure misclassification, however such misclassification should likely be nondifferential. 

## 5. Conclusions

Volunteer firefighters make up over two-thirds of the US fire service, yet are underrepresented in health and exposure studies. PFAS exposure is a growing concern among firefighters and this study provides some evidence that volunteer firefighters have PFAS exposures that differ from those of the general population and from career firefighters. Research on sources of PFAS exposure is an essential focus of future research to inform risk and exposure reduction in firefighters. PFAS are a diverse group of over four thousand compounds, although the vast majority have not been measured in biomonitoring studies. With accumulating evidence of the environmental and health impacts of long chain PFAS, their use in industry and manufacturing, including in AFFF, is being phased out. However, the substitutes often include shorter chain PFAS, the impacts of which are unknown and which are not yet monitored in our public health or environmental monitoring systems. Studies that address the full spectrum of PFAS exposures are needed to understand the full impact of these persistent pollutants.

## Figures and Tables

**Table 1 ijerph-18-03730-t001:** Demographic and firefighting characteristics of CAPS enrollees (n = 135).

Demographics	n (%)	Firefighting Characteristics	n (%)
Age (years)		Years firefighting experience ^1^	
18 to 34	43 (31.8)	0 to 5	36 (26.3)
35 to 49	29 (21.5)	6 to 19	31 (22.6)
50 to 59	27 (20.0)	20 to 34	36 (26.3)
≥60	36 (26.7)	≥35	34 (24.8)
Gender (male)	128 (94.8)	Ever work as a career firefighter ^2^	
Non-Hispanic white	123 (89.8)	Current	9 (6.7)
Education		Former	15 (11.1)
High school graduate	38 (28.1)	Never	111 (82.2)
Some college or Associates degree	63 (46.7)	Average yearly firefighting calls responded to ^3^	
≥4-year college degree	34 (25.2)	0 to 4	42 (35.0)
Usual occupation		5 to 9	25 (20.8)
Construction/manufacturing	45 (33.3)	10 to 19	20 (16.7)
Government/clerical	28 (20.7)	≥20	33 (27.5)
Service provider	15 (11.1)		
Other	47 (34.8)		

^1^ Includes both career and volunteer experience, accounting for any overlapping time between spent at both; ^2^ Career firefighter is defined as a person who works as a firefighter for pay/compensation; ^3^ Calculated as the cumulative number of firefighting calls responded to over the total number of years serving as a firefighter, standardized to one year.

**Table 2 ijerph-18-03730-t002:** PFAS serum concentrations for non-Hispanic white male volunteer firefighters (n = 116) compared with demographically similar members of the U.S. population national averages reported by the National Health and Nutrition Examination Survey (NHANES).

	PFAS Prevalence ^1^			PFAS Serum Levels (ng/mL)							
	CAPS (n = 116)	NHANES		CAPS (n = 116)		NHANES					
		2015–2016 (n = 274)	2017–2018 (n = 272)			2015–2016 (n = 274)			2017–2018 (n = 272)		
				gm ^2^	(95% CI) ^3^	Gm ^2^	(95% CI) ^3^	% diff ^3^	Gm ^2^	(95% CI) ^3^	% diff ^3^
Perfluorononanoic acid (PFNA)	100	98.2	92.1	0.97	(0.89, 1.05)	0.63	(0.56, 0.70)	35.1%	0.46	(0.42, 0.49)	*52.6%*
Perfluorohexanesulfonic acid (PFHxS)	100	98.4	99.4	1.83	(1.61, 2.09)	1.80	(1.55, 2.09)	1.6%	1.70	(1.46, 1.97)	*7.1%*
Perfluorooctanoic acid ^4^ (PFOA)	100	100	100	2.07	(1.89, 2.26)	1.94	(1.76, 2.14)	6.3%	1.74	(1.58, 1.92)	*15.9%*
Perfluorooctanesulfonic acid ^4^ (PFOS)	100	100	100	4.25	(3.76, 4.80)	6.76	(6.13, 7.47)	−59.1%	6.08	(5.44, 6.79)	*−43.1%*
2-(*N*-Methyl-perfluo-rooctane sulfonamido) acetic acid (MeFOSAA)	11.2	38.9	60.6	0.08	(0.07, 0.09)	0.13	(0.11, 0.14)	−62.5%	0.15	(0.12, 0.17)	*−87.5%*
Perfluorodecanoic acid (PFDA)	99.1	69.6	89.3	0.31	(0.29, 0.33)	0.15	(0.13, 0.17)	51.6%	0.19	(0.18, 0.21)	*38.7%*
Perfluoroundecanoic acid (PFUnDA)	46.6	40.8	65.5	0.11	(0.10, 0.12)	0.10	(0.09, 0.11)	9.1%	0.12	(0.11, 0.13)	*−9.1%*
Perfluorododecanoic acid (PFDoA)	80.1	2.4	--	0.14	(0.13, 0.15)	0.07	(0.07, 0.07)	50.0%	Not reported		

^1^ Prevalence = Prevalence was defined as the percent of measurements above the laboratory limit of detection; ^2^ gm = Geometric mean (ng/mL); ^3^ Percent difference between the geometric mean serum levels (ng/mL) of the PFAS compound between participants in CAPS and the respective NHANES cycle; ^4^ NHANES prevalence for these PFAS are unavailable as NHANES measures two separate isomers (linear and branched chain) for PFOA and PFOS rather than one singular compound; these isomers were combined into overall PFOA and PFOS compounds for the purpose of these analyses and NHANES does not measure nor report statistics for an overall PFOA or PFOS compound so prevalence was not presented.

**Table 3 ijerph-18-03730-t003:** Demographic and firefighting characteristics and PFAS serum-concentrations of volunteer firefighters by serum concentrations of selected PFAS, estimated using bivariate analysis.

Characteristics	PFAS Compound (n = 135)					
	PFNA		PFDA		PFDoA	
	Correlation ^1^	*p*-Value ^2^	Correlation ^1^	*p*-Value ^2^	Correlation ^1^	*p*-Value ^2^
Age (years)	0.22	0.011	0.42	<0.001	0.21	0.015
Firefighting experience (years)	0.22	0.011	0.44	<0.001	0.30	0.001
Firefighting calls (yearly n = 120)	0.21	0.023	0.38	<0.001	0.31	0.001
	Geometric mean	(95% CI)	Geometric mean	(95% CI)	Geometric mean	(95% CI)
Occupation						
Construction/manufacturing	0.97	(0.84, 1.10)	0.31	(0.28, 0.35)	0.16	(0.14, 0.18)
Government/clerical	1.08	(0.92, 1.24)	0.33	(0.28, 0.38)	0.15	(0.12, 0.19)
Service provider	0.85	(0.69, 1.00)	0.32	(0.27, 0.37)	0.17	(0.12, 0.21)
Other occupation	1.17	(0.90, 1.44)	0.33	(0.30, 0.35)	0.16	(0.14, 0.18)
*p*-value		0.174		0.685		0.671
Education						
High school graduate	0.96	(0.61, 1.30)	0.27	(0.25, 0.30)	0.14	(0.12, 0.15)
Some college/assoc. degree	1.08	(0.96, 1.21)	0.35	(0.31, 0.38)	0.17	(0.15, 0.19)
≥4-year college degree	1.08	(0.99, 1.17)	0.34	(0.30, 0.37)	0.16	(0.15, 0.19)
*p*-value		0.001		0.003		0.297
Ever a career firefighter						
Yes, currently	1.06	(0.73, 1.38)	0.28	(0.24, 0.32)	0.12	(0.08, 0.15)
Yes, formerly	1.10	(0.96, 1.25)	0.35	(0.31, 0.40)	0.15	(0.12, 0.17)
Never	1.04	(0.90, 1.18)	0.32	(0.30, 0.34)	0.16	(0.15, 0.18)
*p*-value		0.285		0.111		0.139

^1^ Spearman correlation; ^2^ The *p*-value based on the Kruskal–Wallis Chi-Square.

**Table 4 ijerph-18-03730-t004:** Associations between PFAS serum-concentrations and firefighting experience among members of a volunteer fire department, estimated using linear regression (n = 135).

	PFNA			PFDA			PFDoA		
	e^β 1^	95% CI	*p*-Value	e^β 1^	95% CI	*p*-Value	e^β 1^	95% CI	*p*-Value
Firefighting years (per 10 years) ^2^	1.02	(0.93, 1.11)	0.729	1.08	(1.01, 1.15)	0.021	1.19	(1.09, 1.30)	<0.0001
Career firefighter (ref = Current)									
Former	0.91	(0.62, 1.34)	0.642	0.98	(0.74, 1.28)	0.857	0.98	(0.67, 1.45)	0.930
Never	0.93	(0.69, 1.26)	0.650	1.10	(0.89, 1.37)	0.377	1.42	(1.04, 1.93)	0.026
Age (continuous)	1.01	(1.00, 1.01)	0.191	1.00	(1.00, 1.01)	0.359	0.99	(0.99, 1.00)	0.120
Education (ref = High school graduate)									
Some college/assoc.	1.21	(1.02, 1.45)	0.033	1.22	(1.08, 1.39)	0.002	1.19	(1.00, 1.43)	0.055
>4-year college degree	1.29	(1.05, 1.58)	0.017	1.23	(1.06, 1.43)	0.006	1.15	(0.93, 1.42)	0.190
Occupation (ref = Service provider)									
Construction/manufacturing	1.02	(0.79, 1.32)	0.883	0.85	(0.70, 1.02)	0.077	0.85	(0.66, 1.11)	0.235
Government/clerical	1.11	(0.84, 1.47)	0.460	0.86	(0.71, 1.06)	0.156	0.74	(0.56, 0.99)	0.041
Other	1.21	(0.94, 1.56)	0.135	0.99	(0.83, 1.19)	0.918	0.97	(0.75, 1.25)	0.827

^1^ e^β^: exponentiated model coefficients; ^2^ For ease of interpretation, the percent change in the exposure to each PFAS is presented as an increase in 10 years of firefighting experience.

**Table 5 ijerph-18-03730-t005:** Associations between PFAS serum-concentrations and firefighting experience among the subset of active volunteer fire department members who had no current or previous experience as a career firefighter (n = 99), estimated using linear regression.

Characteristic			PFAS Compound						
	PFNA			PFDA			PFDoA		
	e^β^	95% CI	*p*-Value	e^β^	95% CI	*p*-Value	e^β^	95% CI	*p*-Value
Firefighting years (per 10 years)	1.00	(0.91, 1.11)	0.954	1.06	(0.98, 1.15)	0.151	1.17	(1.04, 1.31)	0.008
Firefighting calls/year	1.00	(0.99, 1.01)	0.701	1.00	(1.00, 1.01)	0.403	1.00	(1.00, 1.01)	0.287
Age (continuous)	1.00	(1.00, 1.01)	0.375	1.00	(1.00, 1.01)	0.481	0.99	(0.98, 1.00)	0.219
Education (ref = High school graduate)									
Some college/assoc.	1.20	(0.98, 1.47)	0.074	1.23	(1.04, 1.45)	0.013	1.16	(0.93, 1.45)	0.193
>4-year college degree	1.28	(1.02, 1.61)	0.035	1.23	(1.03, 1.48)	0.026	1.12	(0.87, 1.44)	0.374
Occupation (ref = Service provider)									
Construction/manufacturing	1.08	(0.80, 1.45)	0.614	0.89	(0.70, 1.12)	0.314	1.01	(0.72, 1.40)	0.969
Government/clerical	1.19	(0.86, 1.65)	0.289	0.92	(0.71, 1.19)	0.513	0.92	(0.64, 1.32)	0.651
Other	1.22	(0.91, 1.66)	0.185	1.03	(0.80, 1.31)	0.836	1.11	(0.79, 1.55)	0.545

## Data Availability

Data are associated documentation may be available to users only under a data-sharing agreement that provides for: (1) a commitment to using the data only for research purposes and not to identify any individual participant; (2) a commitment to securing the data using appropriate computer technology; and (3) a commitment to destroying or returning the data after analyses are completed.

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
