# Peer review of "Prevalence and Predictors of Per- and Polyfluoroalkyl Substances (PFAS) Serum Levels among Members of a Suburban US Volunteer Fire Department"

_ijerph, 2021, doi:10.3390/ijerph18073730_

Round 1
Reviewer 1 Report
This biomonitoring study recruited volunteer firefighters to collect blood samples for analysis PFAS. The PFAS distribution and levels among non-Hispanic white adult male study participants were compared to those in the 2015-16 and 2017-18 23 NHANES cycles. The serum levels of PFDoA, perfluorononanoic acid (PFNA), and per- 27 fluorodecanoic acid (PFDA) were elevated among participants compared with NHANES. The levels of both PFDA and PFDoA were positively associated with years of firefighting. The following is the reviewer’s comments:
- This study is well-designed and studied. However, the QA/QC procedures are suggested to be described in brief to make sure that the PFAS levels of each study subject were correct.
- At line 237 to 238,” While 138 members of the volunteer fire department enrolled in the CAPS study, two were missing demographic data and so the study included 136 participants.” The sample number appeared on Table 2 is 116. The authors should explain what led to the differences in sample number.
Author Response
- This study is well-designed and studied. However, the QA/QC procedures are suggested to be described in brief to make sure that the PFAS levels of each study subject were correct.
RESPONSE: We have added information of the laboratory QA/QC procedures for PFAS analysis in the methods section (revised manuscript line 146 to 152)
- At line 237 to 238,” While 138 members of the volunteer fire department enrolled in the CAPS study, two were missing demographic data and so the study included 136 participants.” The sample number appeared on Table 2 is 116. The authors should explain what led to the differences in sample number.
RESPONSE: We understand the reviewer’s confusion. As explained in the methods section, the comparison between CAPS study and NHANES participants was restricted to non-Hispanic white males. We have made this clear by revising the title of the Table 2 to read: ‘PFAS serum concentrations for non-Hispanic white male volunteer firefighters (n=116) compared with demographically similar members of the U.S. population national averages reported by the National Health and Nutrition Examination Survey (NHANES).
Reviewer 2 Report
It is suggested to extend the study follow-up time, as well as to create new associations linked to other health problems such as reproductive health.
Author Response
RESPONSE: We thank the reviewer for this suggestion.
Reviewer 3 Report
The manuscript by Graber et al. describes serum PFAS concentrations in volunteer fire fighters from New Jersey and compares them to historical NHANES data. The Authors were specifically interested in how years of service may correlate to consequent exposure, given that fire protective equipment, gear, and foams have historically contained various PFAS. The authors show that in the study population, PFDoA, PFNA, and PFDA were in higher concentrations as compared to the general population. It is greatly appreciated that the Authors chose to examine how this important group of service workers may be inadvertently exposed to PFAS, and these biomonitoring data could also be important to examine future health effects in this population. However, before publication more methodological details must be added. Additionally, the discussion and a few supporting points could be easily added to enhance the impact of this study.
Major Comments
Line 45 – a reference is needed, and what about inhalation exposure?
Line 50 – I believe these ½ life estimates refer to humans, but this should be stated. Also, PFHxS ½ life estimates are often in the range of >5.5 years depending on the publication.
Line 50 – It would be helpful for the authors to define long chain versus short chain compounds for their context. The definition can vary slightly depending on the context/organization describing them.
Lines 67-69 - What is the point of this sentence/reference? Which were the main PFAS in AFFF?
Line 164 – gender should be switched to sex
Line 191-194 – Please justify/explain why samples below the LOD were assigned this nominal value? In my experience any values below an assay’s LOD/LOQ is assigned a value equal to the LOD.
Line 191-194 – A laboratory’s LLOD and LLOQ will differ based on the instrumentation and assay optimization. It seems odd to me that the Author’s set their LLOD to NHANES, when experimentally their LLOD could be either more or less sensitive, and this should be reported. This comment also ties into the next point below.
Methods – It is appreciated that the Authors reference quite a few publications to describe their methods, and these citations should carry into the next revision(s). However, for the sake of reproducibility and the study’s impact to risk assessment, it is highly suggested that the precise methods for this work are clearly reported so that any other laboratory could readily repeat them. This comment applies more to sample processing and quantification than the data analysis. Reporting standards are important, especially for controversial pollutants like PFAS.
Line 239 - Abstract says average age is 46.6 years (sd. 17.1)
Line ~354 – Some volunteer firefighters travel across country to provide service, especially in times of immediate need like the wildfire crises out West. Could this variable explain some of the trends observed? Given the severity/size of wildfires and the amount of foams and repellants needed to control them, you could hypothesize that these firefighters would have an extremely high exposure to PFAS compared to an average volunteer not involved in wildfire management. Were any volunteers in your cohort involved in wildfire management? Is this somehow reflected in calls/year?
Line 343 – From my current understanding, New Jersey has known contamination of PFDoA in the environment, which may be due to fluorochemical manufacturing in the area. The authors suggest this in line 351, but this point could be expanded upon as it is very interesting. See Goodrow et al. 2020 (doi: 10.1016/j.scitotenv.2020.138839) and Washington et al. 2020 (doi : 10.1126/science.aba7127) amongst others.
Table 2 – In the heading of the table, it reads “PFAS Serum Levels (ug/ml)”. I believe this is a typo and it should be “ng/ml”.
Table 2 – It would be interesting in this table or somewhere else in the publication to report male human half life estimates for each analyte. This would help the reader gauge how years of service may or may affect serum concentrations
Minor Comments
The references should be placed before sentence punctuation, which will help the period splicing throughout the paper.
Line 61 – Please change aka to e.g.,
Line 71 – change “thank” to “than”.
General - Is there standard AFFF/treatment of gear the same across the country?
General - Is there a lot of PFAS contamination at fire stations? How long are volunteers present at the stations? Could this be another source of exposure?
Author Response
Major Comments
- Line 45 – a reference is needed, and what about inhalation exposure?
Response: We thank the reviewer for pointing out this oversite, we have added inhalation as a possible exposure pathway as well as a citation (Poothong, S et al, 2020)
- Line 50 – I believe these ½ life estimates refer to humans, but this should be stated. Also, PFHxS ½ life estimates are often in the range of >5.5 years depending on the publication.
Response: please see the following response (#3 below)
- Line 50 – It would be helpful for the authors to define long chain versus short chain compounds for their context. The definition can vary slightly depending on the context/organization describing them.
Response: We have clarified this information with a more precise summary of the data from the article cited and this sentence now reads: “The serum half-lives of PFHxS, PFOS, and PFOA in humans were reported by one study as 5.3, 3.4, and 2.7 years, respectively”
- Lines 67-69 - What is the point of this sentence/reference? Which were the main PFAS in AFFF?
Response: The sentence was removed as an explanation of physico-chemical transformations of PFAS during active fires is beyond the scope of this paper
- Line 164 – gender should be switched to sex
Response: We thank the reviewer. We collected both sex and gender and the reviewer is correct that this variable captures sex rather than gender.
- Line 191-194 – Please justify/explain why samples below the LOD were assigned this nominal value? In my experience any values below an assay’s LOD/LOQ is assigned a value equal to the LOD.
Response: There are many approaches to LOD imputation, as we explained in the methods section, we chose to use the method that NHANES uses and is widely used in such studies (limit of detection divided by the square root of 2).
- Line 191-194 – A laboratory’s LLOD and LLOQ will differ based on the instrumentation and assay optimization. It seems odd to me that the Author’s set their LLOD to NHANES, when experimentally their LLOD could be either more or less sensitive, and this should be reported. This comment also ties into the next point below.
Response: The DOH laboratory’s LOD were lower than those of NHANES so the NHANES LOD was used for a more accurate comparison, we have clarified this in the manuscript.
- Methods – It is appreciated that the Authors reference quite a few publications to describe their methods, and these citations should carry into the next revision(s). However, for the sake of reproducibility and the study’s impact to risk assessment, it is highly suggested that the precise methods for this work are clearly reported so that any other laboratory could readily repeat them. This comment applies more to sample processing and quantification than the data analysis. Reporting standards are important, especially for controversial pollutants like PFAS.
Response: We very much agree with the reviewer that the laboratory methods must be reproducible. However, we respectfully disagree and believe that accomplishing this by referencing the peer-reviewed articles that describe these methods in detail is acceptable and preferable to readers.
- Line 239 - Abstract says average age is 46.6 years (sd. 17.1)
Response: The abstract is correct and we have made this adjustment in the results section.
- Line ~354 – Some volunteer firefighters travel across country to provide service, especially in times of immediate need like the wildfire crises out West. Could this variable explain some of the trends observed? Given the severity/size of wildfires and the amount of foams and repellants needed to control them, you could hypothesize that these firefighters would have an extremely high exposure to PFAS compared to an average volunteer not involved in wildfire management. Were any volunteers in your cohort involved in wildfire management? Is this somehow reflected in calls/year?
Response: We don’t currently capture this information, but in conversations with the fire department leadership we know that members of this department rarely if ever have worked wildfire management during these crises in the Western US and elsewhere in the world.
- Line 343 – From my current understanding, New Jersey has known contamination of PFDoA in the environment, which may be due to fluorochemical manufacturing in the area. The authors suggest this in line 351, but this point could be expanded upon as it is very interesting. See Goodrow et al. 2020 (doi: 10.1016/j.scitotenv.2020.138839) and Washington et al. 2020 (doi : 10.1126/science.aba7127) amongst others.
Response: We thank the reviewer for bringing the Goodrow et al publication to our attention and have expanded the discussion based on that study’s results (revised manuscript lines 359-366) Interestingly, four of the areas sampled were near our study town. However, among these, PFOA and PFOS were detected in all water and samples but PFDA and PFDoA were not
- Table 2 – In the heading of the table, it reads “PFAS Serum Levels (ug/ml)”. I believe this is a typo and it should be “ng/ml”.
Response: Yes, we thank the reviewer for catching that typo and have corrected it.
- Table 2 – It would be interesting in this table or somewhere else in the publication to report male human half-life estimates for each analyte. This would help the reader gauge how years of service may or may affect serum concentrations.
Response: We agree this is interesting but given that the sources and duration of PFAS exposure are unknown we do not feel it would be informative.
Minor Comments
- The references should be placed before sentence punctuation, which will help the period splicing throughout the paper.
Response: this edit has been made
- Line 71 – change “thank” to “than”.
Response: this edit has been made
- General - Is there standard AFFF/treatment of gear the same across the country?
Response: The variation in PFAS application and exposure through gear is indeed an important question.
- General - Is there a lot of PFAS contamination at fire stations? How long are volunteers present at the stations? Could this be another source of exposure?
Response: An excellent question and one we are exploring.